# Substitutional Coinage Metals as Promising Defects for Adsorption and Detection of Gases on MoS_2_ Monolayers: A Computational Approach

**DOI:** 10.3390/ijms241210284

**Published:** 2023-06-17

**Authors:** Josue Gutierrez-Rodriguez, Miguel Castro, Jose Manuel Nieto-Jalil, Dora Iliana Medina, Saul Montes de Oca, José Andrés García-González, Eduardo Rangel-Cortes, Alan Miralrio

**Affiliations:** 1Escuela de Ingeniería y Ciencias, Tecnologico de Monterrey, Ave. Eugenio Garza Sada 2501, Monterrey 64849, Mexico; a01754207@tec.mx (J.G.-R.); saul.armeaga@tec.mx (S.M.d.O.); anteus79@tec.mx (J.A.G.-G.); 2Departamento de Física y Química Teórica, Division de Estudios de Posgrado, Facultad de Química, Universidad Nacional Autónoma de México (UNAM), Del. Coyoacán, Ciudad de México 04510, Mexico; miguel.castro.m@gmail.com (M.C.); jnietoj@tec.mx (J.M.N.-J.); 3Tecnologico de Monterrey, Institute of Advanced Materials for Sustainable Manufacturing, Monterrey 64849, Mexico; dora.medina@tec.mx

**Keywords:** molybdenum disulfide monolayer, molecular adsorption, half-metallic behavior, greenhouse gases, chemiresistive sensors, magnetic sensors, density functional theory, coinage metal defects

## Abstract

Defective molybdenum disulfide (MoS_2_) monolayers (MLs) modified with coinage metal atoms (Cu, Ag and Au) embedded in sulfur vacancies are studied at a dispersion-corrected density functional level. Atmospheric constituents (H_2_, O_2_ and N_2_) and air pollutants (CO and NO), known as secondary greenhouse gases, are adsorbed on up to two atoms embedded into sulfur vacancies in MoS_2_ MLs. The adsorption energies suggest that the NO (1.44 eV) and CO (1.24 eV) are chemisorbed more strongly than O_2_ (1.07 eV) and N_2_ (0.66 eV) on the ML with a cooper atom substituting for a sulfur atom. Therefore, the adsorption of N_2_ and O_2_ does not compete with NO or CO adsorption. Besides, NO adsorbed on embedded Cu creates a new level in the band gap. In addition, it was found that the CO molecule could directly react with the pre-adsorbed O_2_ molecule on a Cu atom, forming the complex OOCO, via the Eley–Rideal reaction mechanism. The adsorption energies of CO, NO and O_2_ on *Au_2S2_*, *Cu_2S2_* and *Ag_2S2_* embedded into two sulfur vacancies were competitive. Charge transference occurs from the defective MoS_2_ ML to the adsorbed molecules, oxidizing the later ones (NO, CO and O_2_) since they act as acceptors. The total and projected density of states reveal that a MoS_2_ ML modified with copper, gold and silver dimers could be used to design electronic or magnetic devices for sensing applications in the adsorption of NO, CO and O_2_ molecules. Moreover, NO and O_2_ molecules adsorbed on MoS_2_-*Au_2_s_2_* and MoS_2_-*Cu_2_s_2_* introduce a transition from metallic to half-metallic behavior for applications in spintronics. These modified monolayers are expected to exhibit chemiresistive behavior, meaning their electrical resistance changes in response to the presence of NO molecules. This property makes them suitable for detecting and measuring NO concentrations. Also, modified materials with half-metal behavior could be beneficial for spintronic devices, particularly those that require spin-polarized currents.

## 1. Introduction

The potential applications, such as optoelectronics, of molybdenum disulfide (MoS_2_) monolayer (ML) have been widely studied due to its optical gap ranging from 1.8 to 1.9 eV [1] and high thermal stability [1], as well as potential applications in: hydrogen storage [2], hydrodesulfurization processes in crude oil [3], sensors [4,5,6,7], and hydrogen evolution reaction (HER) [8]. Conversely, the chemically inert surface of the MoS_2_ ML hinders using it in other ways. Substitutional atoms at Mo or S sites [9], the formation of vacancies [10,11,12], edge defects [13] and usage of different substrates [14] have been used to chemically activate the ML. Experimentally, the electron irradiation technique can be used to create a single S in a controlled manner [15]. In addition, sulfur vacancies have been proven to introduce new states in the band gap, mostly from the contribution of Mo (*d)* orbitals, progressively reducing the energy gap [16].

Inspired by these results, our research team studied defective MoS_2_ MLs, with up to two coinage metal (Cu, Ag, and Au) atoms embedded into S vacancies [16,17]. Formation energies, calculated for each sulfur vacancy *V_nSn_* (*n* = 1–3) average 2.85 eV. Thus, vacancy formation is a manner to chemically activate the previously inert ML [16,17]. In addition, substitutional atoms also create defect states in the bandgap of pristine ML. The chemical activation of the MoS_2_ ML surface using these embedded metal atoms was confirmed through Fukui reactivity indices [16,17].

Sharma et. al., studied the energetic stability of a copper atom in the sulfur defective site of the MoS_2_ ML via the reaction path from S vacancy to neighboring sites [18]. They found that the minimum energy barrier for this migration is 1.37 eV, whereas the reaction energy is 1.11 eV. Thus, this reaction is endothermic showing that the copper atom adsorbed is quite stable and can stably disperse on MoS_2_ sheets. Besides, our results show that Au, Ag and Cu metal atoms embedded in S defect sites in MoS_2_ ML are stable [16,17]. Similarly, Zhao et al. studied the electronic properties of MoS_2_ doped with Re, revealing spin polarization in composed systems [19]. Besides, Fan et al. studied the structures of several transition metal atoms embedded in MoS_2_ MLs [20]. Strong interactions suggest that several embedded atoms on the MoS_2_ ML are stable. Also, Tong et al. provided insights on the hydrogen oxidation reaction on MoS_2_ nanosheets modified with 3*d* transition metal atoms [21].

Thus, the adsorption of small molecules, such as those composing the atmosphere (H_2_, O_2_, N_2_) or small atmospheric pollutants, might be possible by the enhanced reactivity achieved by substitutional metal atoms on MoS_2_ MLs, in comparison with the pristine MoS_2_ basal plane. Since the production of carbon monoxide (CO), as well as nitric oxide, (NO) is mainly anthropogenic [22], being known as secondary greenhouse gases, modified MoS_2_ MLs can be proposed to reduce their levels by trapping them. Currently, according to sustainable development goals promoted by the United Nations, the reduction of greenhouse gases is a global priority [23]. Also, defective electronic levels yielded by the substitutional atoms on MoS_2_ could lead to ultrasensitive sensors of pollutants in the gas phase [24,25]. Similarly, the chemisorption of molecules could introduce changes in the electronic and magnetic behavior of the layer [26]. Also, the modified energy landscape can lead to selective adsorption and further separation of gas mixtures, such as CH_4_ and N_2_ molecules, recently reported by Wu et al. [27]. Thus, novel strategies to trap or detect small greenhouse gases, such as CO or NO, on modified MoS_2_ MLs are worth studying.

If the ML is exposed to reactive species of air, N_2_ and O_2_, those could be strong inhibitor competitors for the CO and NO adsorption, or these molecules could even react with them, forming other species. For example, the oxidation of CO catalyzed by the MoS_2_ ML with a *Cu_S_* atom embedded in the S vacancy was studied recently [28]. Also, it was reported that O_2_ interacts more strongly than the CO molecule on the *Cu_S_* atom, with binding energies of about 2.115 and 1.249 eV, respectively. Subsequently, O_2_ should cover the copper atom and assist the oxidation of CO. Those CO and O_2_ adsorption energies also were calculated in another study [20]. The O_2_ and CO adsorption energies values found were 1.16 and 1.27 eV, respectively. Analogously, other authors reported the adsorption energy of the O_2_ molecule on the MoS_2_-*Cu_S_* monolayer as 0.99 eV [24]. These last results suggest that the CO molecule adsorption could compete with the O_2_ adsorption on the MoS_2_-*Cu_S_* system. So, the oxidation of CO might not be as efficient. That is, the interaction of diatomic molecules on molybdenum disulfide MLs with Cu, Ag and Au atoms embedded into S vacancies are of potential technological relevance by their catalytic, electronic and magnetic properties introduced by the molecular adsorption. Furthermore, the study of the interaction of the MoS_2_ surfaces with the O_2_ molecule is very relevant, since the small amount in the gas phase could give way to Cu, Au and Ag oxidation.

Several devices have already been manufactured to detect gases and semiconductor/chemiresistive sensors have been adopted commercially owing to their low-cost and straightforward fabrication process [29]. Besides, chemiresistive sensors based on tungsten and disulfide, freestanding as well as modified with zinc oxide nanoparticles, were fabricated and reported [30]. These systems showed a significant response to a range of target analytes. The sensing material has an inherent resistance, which can be modulated by the presence or absence of the analyte, leading to the chemiresistive behavior. In addition, the hydrodesulfurization process, crucial to remove sulfur from petroleum and produce ultraclean fuels, commonly employs derivatives of bulk MoS_2_ as a catalyst [31,32]. Recently, the interest of the petrochemical industry in materials with improved performance has attracted attention to nanostructured forms of MoS_2_, such as nanoparticles [33], nanorods and nanosheets [34,35].

These materials based on nanostructured MoS_2_ are not only relevant for sensing applications [7,26,36,37], but also hold potential as catalysts for petrochemical applications. Utilizing nanostructured forms of MoS_2_ is a feasible approach considering the widely used applications of its common bulk derivatives.

More rare and peculiar would be a sensing material that has an inherent transition from metallic to half-metallic behavior, which can be modulated by the presence or absence of molecules. In this manuscript, a few findings on half-metallic materials for gas sensing are discussed. However, further improvements are needed in the potential manufacturability of sensors based on 2D materials such as molybdenum and tungsten disulfides for their practical applications.

Thus, the current work aims to study the adsorption of diatomic molecules AB (AB = H_2_, O_2_, N_2_, CO and NO) on group 11 atoms M (M = Cu, Ag and Au) embedded into mono- (*V_S_*) and di- vacancies (*V_2S2_*) of sulfur on MoS_2_ MLs. For the first time, this study attempts to contribute to the understanding of the electronic behavior of diatomic molecules adsorbed on *Au_2S2_*, *Cu_2S2_* and *Ag_2S2_* dimers embedded into two sulfur vacancies by means of dispersion-corrected density functional theory (DFT). In addition, half-metallic systems with intrinsic ferromagnetism were constructed and identified, making them promising materials for spintronics and sensing applications.

## 2. Results

Since MoS_2_ ML is considered inert, introducing defects creates states in the band gap. Thus, modified surfaces are more prone to interact with other species [16,17]. Metal atoms *M_S_* (M = Cu, Ag or Au) embedded into S vacancies are found pushed above the atomic layer (Figure 1). 

The favorable binding energies calculated for metal atoms embedded into sulfur vacancies, of about 3.33 (*Au_S_*), 2.63 (*Ag_S_*); and 3.19 eV (*Cu_S_*), 2.40 (*Au_2S2_*), 1.82 (*Ag_2S2_*), 2.165 eV/atom (*Cu_2S2_*), suggest that the substitutional atoms remain attached to the correspondent vacancies. Likewise, defects form new levels in the band gap of the ML.

These peaks are obtained for Mo (*d*) and *p* orbitals of substitutional atoms. Just below the valence band minimum of all monolayers, levels are formed by Mo (*d*) levels. In contrast, over the conduction band, minimum orbital *s* contributions also appear from substitutional atoms. Band gaps of 0.938, 0.841, 0.948 and 1.102 eV were found for MoS_2_-*V_S_*, MoS_2_-*Cu_S_*, MoS_2_-*Ag_S_* and MoS_2_-*Au_S_*, respectively. Besides, band gaps of 0.799, 0.538, 0.642, and 0.785 eV were found for MoS_2_-*V_2S2_*, MoS_2_-*Cu_2S2_*, MoS_2_-*Ag_2S2_* and MoS_2_-*Au_2S2_*, respectively. These levels also introduced band gap changes relative to the pristine ML. All the above calculations are in full agreement with our previous reports [16,17].

All non-equivalent conformations were initially considered, but only the ground state structures were studied in this work. Subsequently, atmospheric components and some air pollutants interacting with the defective surfaces MoS_2_-*M_S_* (M = Cu, Ag or Au) were fully optimized in several conformations. Then, it was found that diatomic molecules were attached to the preferential interaction region introduced by the substitutional metal atoms (Figure 1 and Figure 2). The above configurations were obtained in the case of these systems with a single defect (Table 1).

In contrast, pristine MLs do not form stable structures when atmospheric components and air pollutants attach to their surfaces. Thus, the chemical activation of the molybdenum disulfide ML is confirmed by substituting metal atoms into sulfur vacancies. Adsorption of diatomic molecules is energetically favorable in most cases, with higher adsorption energies for O_2_, CO, and NO attached to the embedded copper atom (Table 1). The shortest distances between the copper atom and the CO and NO molecules are 1.86 and 1.78 Å (Figure 2), respectively. Moreover, larger interatomic bond lengths were obtained for the CO and NO molecules relative to their values in the gas phase, as evidenced by the interaction between these diatomic molecules and the copper atom. In these cases, CO and NO molecules are attached to the defective monolayer by C and N atoms, correspondingly (Table 1). Interatomic bond lengths for CO and NO on MoS_2_-*Cu_s_* are 1.14 and 1.18 Å, respectively. In comparison, in the gas phase, these bond lengths are 1.12 and 1.15 Å, respectively.

Similarly, an adsorbed O_2_ molecule is obtained with a longer O-O bond length, relative to 1.23 Å computed in the gas phase. The 1.32 and 1.34 Å O-O bond lengths obtained in cases of MoS_2_-*Ag_s_* and MoS_2_-*Cu_s_* suggest the achievement of the superoxide state in the case of the oxygen molecule [38]. This interaction leads to atop adsorption modes (Table 1), with the shortest distance between the defect and O_2_ molecule ranging from 1.97 to 2.27 Å (Figure 1).

Thus, the adsorption energies indicate that NO and CO are more strongly chemisorbed than O_2_ and N_2_ on the ML when a copper atom substitutes for a sulfur atom (Table 1). The above can be visualized by the shorter bond lengths between NO (1.78 Å) and CO (1.86 Å) molecules and defective surface MoS_2_-*Cu_S_*, in comparison with those established with O_2_ (1.97 Å) and N_2_ (1.91 Å) (Figure 1 and Figure 2). Therefore, N_2_ and O_2_ adsorption could not compete at all with that of NO or CO molecules. However, when the bidimensional MoS_2_ with silver and gold atoms embedded into sulfur vacancies is exposed to species of air (H_2_, N_2_ and O_2_), the results suggest that these molecules could act as strong inhibitors, competing with CO and NO adsorption.

The enlargement observed in the air pollutants is attributed to charge transference occurring in the defective MoS_2_ MLs. This charge transfer can facilitate the detection of these molecules by occupying the defective states and inducing additional changes in the MoS_2_ electronic or electric transport properties. The density of states, see Figure 3, show that the adsorption of the CO molecule does not result in any obvious change in the MoS_2_ electronic properties (Appendix A). Contrastingly, NO and O_2_ adsorbed on substitutional Cu creates a new level in the band gap of the MoS_2_ ML (Figure 3). Likewise, PDOS for Mo, S, *Cu_S_*, N and O atoms suggest O(*p*)-N(*s*)-Cu(*p*) hybridization for the new level below the Fermi level, and N(*s*)-Cu(*d*) and Mo(*d*)-Cu(*p*) contributions for defective states above the Fermi energy, with spin-up and spin-down, respectively. Magnetic moments per cell are obtained as 0.7 and 2.07 μ_B_, for MoS_2_-*Cu_S_* and MoS_2_-*Cu_S_*-NO, respectively (Appendix A). Moreover, Bader analysis shows charge transference from the embedded Cu atom to NO and O_2_ molecules (Table 1). Thus, these behaviors are promising for detecting air pollutants by electronic or magnetic sensing devices based on defective molybdenum disulfide.

It has been found that the O_2_ molecule does not react with the pre-adsorbed CO or NO molecules (Figure 4), nor displace them. Contrastingly, the CO molecule directly reacts with the pre-adsorbed O_2_ molecule on atom Cu, forming the complex OOCO via Eley–Rideal (ER) reaction mechanisms (Figure 4). In two works [28,39], the authors reported two different mechanisms for the reactions of the CO oxidation on pre-adsorbed O_2_ on MoS_2_-*Cu_S_*: ER and Langmuir–Hinshelwood (LH) mechanisms, respectively. Therefore, the reaction process could occur in two stages: the LH reaction, CO + O_2_ → OOCO, in which carbon monoxide is oxidized and leads to the next second step: OOCO + CO → 2CO_2_ via the ER process — thus, resulting in a couple of CO_2_ molecules without any energy barrier [28]. Our results are consistent with the report [39], which through ER mechanisms obtained that the firstly adsorbed CO molecule reacts directly with an activated O_2_ molecule on the Cu_4_ cluster embedded on sulfur tetra-vacancy on MoS_2_, with a barrier energy of 0.385 eV. Thus, the reactions can occur rapidly under room conditions, and therefore, the system is suitable for efficient recovery.

There is no experimental or theoretical evidence for OONO + NO → 2NO_2_ on MoS_2_-*Cu_S_* via ER reaction mechanisms. It has been found that the NO molecule is adsorbed on the pre-adsorbed O_2_ molecule without dissociation. Also, the NO adsorption energy on Cu-O_2_ was of −0.854 eV. Besides, NO oxidizes easily with the O_2_ in air (2NO + O_2_ → 2NO_2_), leading to abundant nitrogen dioxide in the atmosphere. The activation energy of the reaction 2NO + O_2_ was reported previously [40]. The finest estimation of the activation energy was 6.47 kJ/mol (70 meV). Thus, it is noticeable that the modified MoS_2_ ML system doped with an embedded Cu atom is highly appropriate for absorbing NO_2_ molecules as in the case of the NO molecule, in full agreement with the work of Zhao et al. [24].

Therefore, modified MoS_2_ ML with *Au_2S2_*, *Ag_2S2_* and *Cu_2S2_* dimers trapped in di-sulfur vacancies to improve the adsorption atmospheric components (H_2_, O_2_ and N_2_) and secondary greenhouse gases (CO and NO) has been theoretically proven for the first time (Figure 5 and Figure 6). Properties induced by defects under study imply that embedded *Au_2S2_* and *Cu_2S2_* defects are promising materials for sensing purposes for the adsorption of the NO and CO molecules. 

Adsorption of diatomic molecules is energetically favorable in most cases, with higher adsorption energies for O_2_, CO and NO attached to the embedded *Cu_2S2_*, *Au_2S2_* and *Ag_2S2_* dimers (Table 2). The adsorption energies show that NO and CO are chemisorbed stronger than O_2_ and N_2_ on the ML with *Au_2S2_* atoms substituting two sulfur atoms.

The adsorption energies of O_2_, CO and NO on *Ag_2S2_* and *Cu_2S2_* dimers embedded into sulfur vacancies are similar. In contrast, O_2_ is the only species that forms bridge adsorption modes with the pairs of embedded metal atoms (Table 2). The other species are shown above in the adsorption modes, firmly attached to a single substitutional defect on the heteroatoms (Table 2).

Furthermore, it has been found that O_2_-NO and O_2_-CO pairs of molecules could be adsorbed on *Au_2S2_*, *Cu_2S2_* and *Ag_2S2_* (Figure 6). All possible reactions among two or three molecules on Au_2_, Ag_2_ and Au_2_ clusters embedded into *V_2S2_* defects are more complicated processes. Therefore, it is proposed that these pairs be studied in future reports. The shortest distances between the *Au_2S2_* dimer and CO and NO molecules are 2.00 (C-Au) and 2.17 (Au-N) Å, respectively (Figure 6). The most suitable adsorption mode of CO shows the C atom bonded with embedded gold and with Au-C bonded perpendicular to the plane. The NO molecule forms an Au-N-Au bridge perpendicular to the plane and O_2_ forms an Au-O-O-Au bridge. Moreover, larger interatomic bond lengths were obtained for the CO and NO molecules relative to their values in the gas phase, as evidenced by the interaction between these diatomic molecules and the Au_2_ dimer. Interatomic bond lengths for CO and NO on MoS_2_-*Au_2s2_* are computed as 1.15 and 1.20 Å, respectively. In comparison, in the gas phase, these bond lengths are only of about 1.12 and 1.15 Å, respectively.

Charge transference occurs from the defective MoS_2_ ML to the adsorbed molecules, oxidizing the later ones (NO, CO and O_2_) since they act as acceptors (Table 2). This effect can be explained as follows: vertical electronic affinities (EA) for MoS_2_-Au_2S2_, MoS_2_-Cu_2S2_ and MoS_2_-Ag_2S2_ are computed as 0.22, 0.37 and 0.170 eV, respectively. In comparison, vertical ionization potentials (IP) for the same systems are obtained as 1.71, 1.91 and 1.68 eV, respectively. In contrast, EA/IP values for O_2_, NO and CO are 1.08/12.63, 1.17/9.92 and 2.68/13.82 eV, respectively. Thus, as a result, it is difficult to remove electrons from the adsorbed molecules. Also, their electron affinities are larger than those obtained in the case of the modified MLs, leading to the tendency to donate charge from the defective material to the adsorbed molecules. Similarly, the fraction of electrons transferred (∆N) can be related to the charge donated. This amount was computed as ∆N = (χ_ML_ − χ_AB_)·[2(η_ML_ + η_AB_)]^−1^, in which χ = (IP + EA)/2 denotes absolute electronegativity, whereas η = (IP − EA)/2 the absolute hardness, for the correspondent ML and diatomic molecule, respectively [41]. These results are consistent with the flux of charge coming from the defective ML and adsorbed molecule, as stated in Table 2. DOS and PDOS, Figure 7, show that the CO adsorption molecule results in a change in the electronic properties of MoS_2_-Au_2S2_. The DOS peak of spin-up-down at the conduction band minimum (CBM) is disappearing for MoS_2_-Au_2S2_. Likewise, PDOS for Mo, S, Au_S_, C and O atoms suggest O(p)-C(s,p)-Au(s,d) hybridization at the Fermi level and above the Fermi level for spin-up and spin-down, respectively. The magnetic moments per cell are obtained at 0.06 and 0.03 μB, for MoS_2_-Au_2S2_ and MoS_2_-Au_2S2_-CO, respectively. The system MoS_2_-Cu_2S2_-CO has a similar behavior in DOS and PDOS to the system MoS_2_-Au_2S2_-CO (Figure 7). For the MoS_2_-Ag_2S2_ and MoS_2_-Ag_2S2_-CO systems, no changes are observed in DOS and PDOS at the Fermi level (Figure 7).

Contrastingly, NO adsorbed on substitutional Au_2_ creates a new level in the band gap of the MoS_2_-Au_2S2_ ML (Figure 7), which induces spin polarization by unpaired electrons, shown in the spin density maps, and a total magnetic moment of about 0.87 μ_B_. The peak of spin-up at the Fermi level disappears, whereas new states are shown at the valence band maximum (VBM) of the MoS_2_-Cu_S_-NO system. Besides, the MoS_2_-Au_2S2_-NO ML has a half-metallic and ferromagnetic behavior, with a band gap of about 0.5 eV for spin-up (Figure 7). The transition from metal to half-metal for spin-up electrons is mainly related to the hybridization of s and d orbitals of Au atoms and the *s*, *p* orbitals of N and O atoms in the MoS_2_-*Au_2S2_*-NO system, respectively, as shown in Figure 7. Bader charge analysis exhibits that, 0.52 e are transferred from the ML to the NO. On the other hand, the NO molecule adsorbed on MoS_2-_*Cu_2S2_* introduces a transition from metallic to semiconductor behavior for spin-up-down electrons as shown in Figure 7. Also, the MoS_2_-Ag_2S2_ system has a similar behavior to MoS_2_-*Cu_2S2_*.

Since O_2_ leads to peculiar behaviors in the case of single substitutional defects, the following discussion elucidates its interaction with embedded dimers. First, it is noticed that O_2_, adsorbed on any dimer under study achieves the superoxide state. The above can be argued by the O-O bond lengths obtained, ranging from 1.34 to 1.38 Å (Figure 5 and Figure 6) [38]. In addition, the total magnetization of these systems is obtained ranging from 0.26 to 0.94 μ_B_ (see Table 2). However, this total magnetization is comparable to the individual contribution of each oxygen atom adsorbed on *Au_2S2_* or *Cu_2S2_* defects (Table 2). Since this fact sounds somewhat contradictory, spin density isosurfaces were obtained (Appendix A). It is noticed that uniquely in the case of O_2_, obtained with important spin-up contributions, total magnetization is reduced by the introduction of spin-down electrons on Mo atoms located behind the embedded defect. It is relevant to note that the Mo layer acts as an electron reservoir, transferring a certain amount of charge to the adsorbed molecule through the metal defect (Appendix A). Thus, it resembles a parallel plate capacitor discharging into the adsorbed molecule as a resistive component.

As discussed below, the charge transferred from the ML to the O_2_ molecule induced its superoxide state. Admirably, at the same time, antiparallel electrons are found surrounding Mo atoms near *Au_2S2_* or *Cu_2S2_* defects. The above is consistent with recent reports on the formation of reactive oxygen species on nanostructured MoS_2_ derivatives [42,43].

Finally, it is worth mentioning that the O_2_ molecule adsorbed on embedded *Au_2S2_* or *Cu_2S2_* induces a transition from metal to half-metal for spin-up electrons, which is mostly related to the hybridization of s and d orbitals of Au or Cu and the *s*, *p* orbitals of O atoms in the MoS_2_-*Au_2s2_*-O_2_ and MoS_2_-*Cu_2s2_*-O_2_ system, respectively, as shown in Figure 7. The bandgaps are approximately of 0.7 and 0.5 eV, for MoS_2_-*Au_2S2_*-O_2_ and MoS_2_-*Cu_2S2_*-O_2_ systems, respectively. Bader charge analysis shows that about 0.71 and 0.85 e are transferred from the MoS_2_-*Au_2S2_* and MoS_2_-*Cu_2S2_* systems to O_2_ molecule, respectively (Table 2). Contrastingly, O_2_ adsorbed on *Ag_2S2_* embedded dimer creates a new level in the energy gap of MoS_2_-*Ag_2S2_* ML (Figure 7), which introduces an excess of unpaired electrons, shown in the spin density maps and total magnetization, of about 0.94 μ_B_ (See Table 2). In this case, no half-metallic behavior is observed. Curiously, it has been found that CO or NO molecules do not react with the pre-adsorbed O_2_ molecule and do not displace it. Appendix A shows the electrostatic potential surface for all systems. In the case of O_2_ adsorbed on MLs modified with embedded metal dimmers, the EPS maps reveal electron-rich regions, red-colored, above the adsorbed molecule. Thus, the oxygen molecule shields the electron-poor regions, blue-colored, obtained behind. Finally, the H_2_ and N_2_ molecules adsorption does not lead to relevant changes in the electronic structure of MLs modified with coinage metal dimers embedded into sulfur vacancies (Appendix A).

Overall, MoS_2_-*M_S_* and MoS_2_-*M_2S2_* systems possess intrinsic electrons and holes at a temperature greater than absolute zero. In the absence of an interacting gas, such as O_2_, CO or NO, the sensor would ideally show a given conductivity and resistivity. However, the charge distribution in the sensing material could change when it interacts with an external gas molecule, and the modification depends upon the strength of the molecule-ML chemical interaction. Strong interactions between NO and O_2_ adsorbed on the MoS_2_-*Cu_S_* system is attributed to the charge transfer and orbital hybridization between the molecules and embedded *Cu_S_* atom. Thus, MoS_2_-*Cu_S_*, proposed as chemiresistive sensing material, showed a significantly higher response to NO. Upon adsorption, each spin-polarized 2π* level of the NO is splitting into two states and in the spin up 2π*, the level is shifted below the Fermi level due to the hybridization between the Cu (*d*) and NO (2π*) orbital.

On the other hand, a more peculiar behavior is shown by MoS_2_-*Au_2S2_* and MoS_2_-*Cu_2S2_* sensing materials. The above is due to their transition from metallic to half-metallic materials, which can be modulated by the presence or absence of O_2_ and NO. Once again, spin-up 2π* orbitals, for NO or O_2_, are shifted below the Fermi level due to the hybridization between *d* metal orbitals and 2π* ones of the interacting molecules, thus an energy gap is opened.

## 3. Methods and Materials 

The dispersion-corrected density functional method, labeled as PBE-D3(BJ)/USP, was previously used to obtain reliable results for defective MoS_2_ ML with substitional coinage metals [16,17]. The quantum chemistry package Quantum Espresso 6.1 [44] was used for most of the calculations. The full methodology of ultra-soft pseudopotentials, obtained by the Rappe Rabe Kaxiras Joannopoulos method, was used for the interacting molecules as well. Furthermore, in previous studies, it was demonstrated that the 4 × 4 × 1 supercell is sufficiently large to accommodate the molecules under investigation (H_2_, N_2_, O_2_, NO and CO). Previously, a 5 × 5 × 1 supercell was tested at the PBE-D2/USP level [16], obtained no relevant geometrical differences in comparison to the smaller one.

The energy was converged up to 1.0 × 10^−7^ a.u. with an energy cut-off of 60 Ry. The convergence criteria for the geometry optimization considered a threshold on the total energy of 1.0 × 10^−5^ a.u., and 1.0 × 10^−4^ a.u., for the residual forces on each atom. Besides, the BFGS quasi-Newton algorithm was used for structural relaxation. The Brillouin zone was sampled through a 4 × 4 × 1 k-point grid. The mesh was increased up to 16 × 16 × 1 for the density of states and the projected density of states calculations.

The lattice parameter obtained, after a full relaxation of the unit cell, was 3.199 Å. This value is pretty similar to the 3.169 Å value measured for the bulk phase MoS_2_ [45]. Another value of 3.16 Å was obtained, via scanning tunneling microscopy, on MoS_2_ monolayers on highly ordered pyrolytic graphite [46].

On the other hand, a defect *X_y_* refers to X species, such as metal atoms M_n_ (*n* = 1, 2) or vacancies V_n_ (*n* = 1, 2), occupying the site of the Y species into MoS_2_ ML; sulfur atoms S_n_ (*n* = 1, 2) in the current case. Multiple sulfur vacancies are denoted as *V_Sn_* (*n* = 1, 2). Embedded clusters *M_nSn_* (*n* = 1, 2, M = Cu, Ag or Au) occupying sulfur vacancies were considered. The adsorption energies of the metal atoms embedded into sulfur vacancies and the adsorption energies of the diatomic molecules on metal were calculated as follows:
E_Metal-ads_ = E(MoS_2_ − *V_nSn_*) + E(*M_n_*) − E(MoS_2_ − *M_nSn_*)(1)
E_AB-ads_ = E(MoS_2_ − *M_nSn_*) + E(AB) − E(MoS_2_ − *M_nSn_* − AB)
(2)
E_AB-AB-ads_ = E(MoS_2_ − *M_nSn_* + AB) + E(AB) − E(MoS_2_ − *M_nSn_* − AB − AB)(3)

E(MoS_2_ − *V_nSn_* + AB), E(MoS_2_ − *M_nSn_*) and E(MoS_2_ − *M_nSn_* − AB) are the total energies of defective MLs: with *n* sulfur vacancies *V_nSn_*, with embedded clusters *M_nSn_* (*n* = 1, 2, M = Cu, Ag or Au) occupying sulfur vacancies, and interacting with AB diatomic molecules, respectively. Structures with two metal atoms to form the defects *M_nSn_* (*n* = 1–2) were studied. Total electrostatic potential (ESP) was mapped on isosurfaces with 0.01 a.u. of electron density. Bader’s charge analyses were calculated by Henkelman’s group software version 1.04 [47]. Similarly, individual contributions of the A and B atoms, forming the AB molecule, to the total magnetization were computed by the Bader’s procedure and spin density.

## 4. Conclusions

Chemical activation of the molybdenum disulfide ML was achieved through coinage metal atoms (Cu, Ag and Au) embedded into mono- and di-sulfur vacancies *V_S_*_,_ and *V_2S2_*, respectively.

Atmospheric components (N_2_, O_2_, H_2_) and air pollutants known as secondary greenhouse gases (CO and NO) can be attached to the surface since their adsorptions are energetically favorable in most cases. Among them, CO and NO on the substitutional copper atom obtained higher adsorption energies. Furthermore, N_2_ and O_2_ adsorption could not compete with NO or CO adsorption at all.

The enlargement of bond lengths observed after adsorption serves as preliminary evidence of bond weakening due to charge transfer from the copper atom embedded in the sulfur vacancies of the CO and NO molecules.

The adsorption of the CO, N_2_ and H_2_ molecules does not lead to changes in the electronic properties of the defective MoS_2_ MLs. In contrast, as evidenced by total and projected density of states, NO adsorbed on embedded Cu creates a new level in the band gap of the MoS_2_ ML.

In addition, it was found that the CO molecule could directly react with the pre-adsorbed O_2_ molecule on the Cu atom, forming the complex OOCO, via the Eley–Rideal reaction mechanism.

The adsorption energies of CO, NO and O_2_ on *Au_2S2_*, *Cu_2S2_* and *Ag_2S2_* dimers embedded into sulfur di-vacancies were similar. Charge transfer occurs from the modified MLs to the adsorbed molecules, oxidizing NO, CO and O_2_ and acting as acceptors.

In particular, NO and O_2_ molecules adsorbed on MoS_2_-*Au_2S2_* and MoS_2_-*Cu_2S2_* introduces a transition from metallic to half-metallic behavior. Interestingly, CO and NO molecules do not react with the pre-adsorbed O_2_ molecule. As a result, it does not displace it.

On the other hand, the adsorption of NO molecules onto MoS_2_-*Cu_2S2_* induces a transition from metallic to semiconductor behavior, for spin-up-down electrons. Thus, this material is interesting for spintronic applications.

For the first time, the electronic behavior of some diatomic molecules adsorbed on *Au_2S2_*, *Cu_2S2_* and *Ag_2S2_* embedded into sulfur di-vacancies was studied as well. The introduction of the diatomic molecules affects mostly electronic and magnetic properties.

The changes induced by the adsorbed molecules are selective, meaning they depend on the specific chemical species involved. This observation suggests potential applications for the modified MLs as gas phase molecule sensors.

Sensors based on modified MoS_2_ monolayers, due to their expected chemiresistive behavior, can be proposed for NO molecules. Furthermore, the introduction of certain gas molecules has been found to modify the magnetic properties of the materials, resulting in a new category of sensitive materials. Additionally, the modified materials exhibit an intriguing phenomenon known as half-metal behavior, which holds potential for spintronic devices that rely on spin-polarized currents.

## Figures and Tables

**Figure 1 ijms-24-10284-f001:**
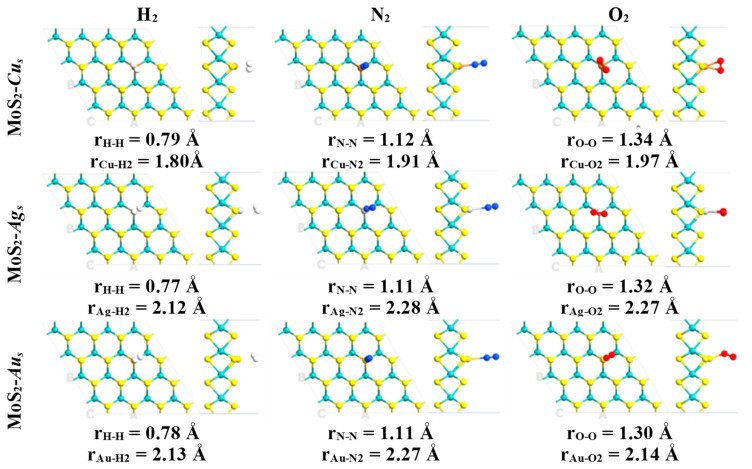
Ground state structures obtained for defective MoS_2_-*M_S_* (M = Cu, Ag and Au) monolayers interacting with the atmospheric constituents (H_2_, N_2_, and O_2_). Relevant bond lengths are indicated in angstrom units. Mo, S, H, N and O atoms are indicated with cyan, yellow, white, blue and red colors, respectively.

**Figure 2 ijms-24-10284-f002:**
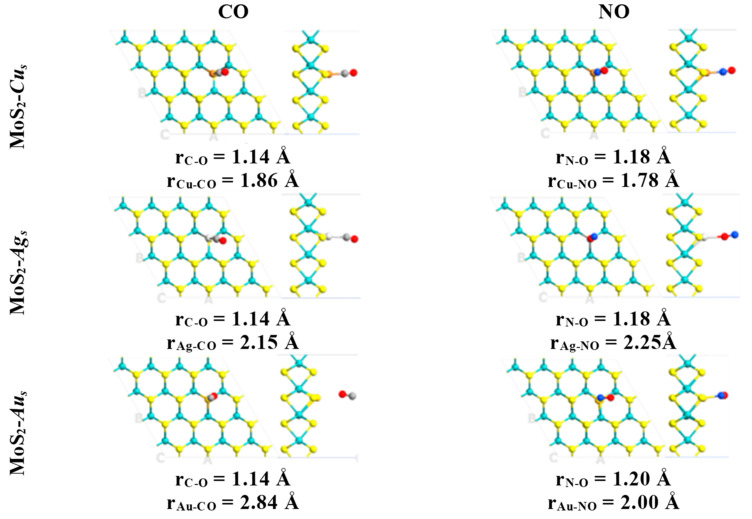
Ground state structures obtained for the defective monolayers MoS_2_-*M_S_* (M = Cu, Ag and Au) interacting with air pollutants CO and NO. Relevant bond lengths are indicated in angstrom units. Mo, S, C, N and O atoms are indicated with cyan, yellow, gray, blue and red colors, respectively.

**Figure 3 ijms-24-10284-f003:**
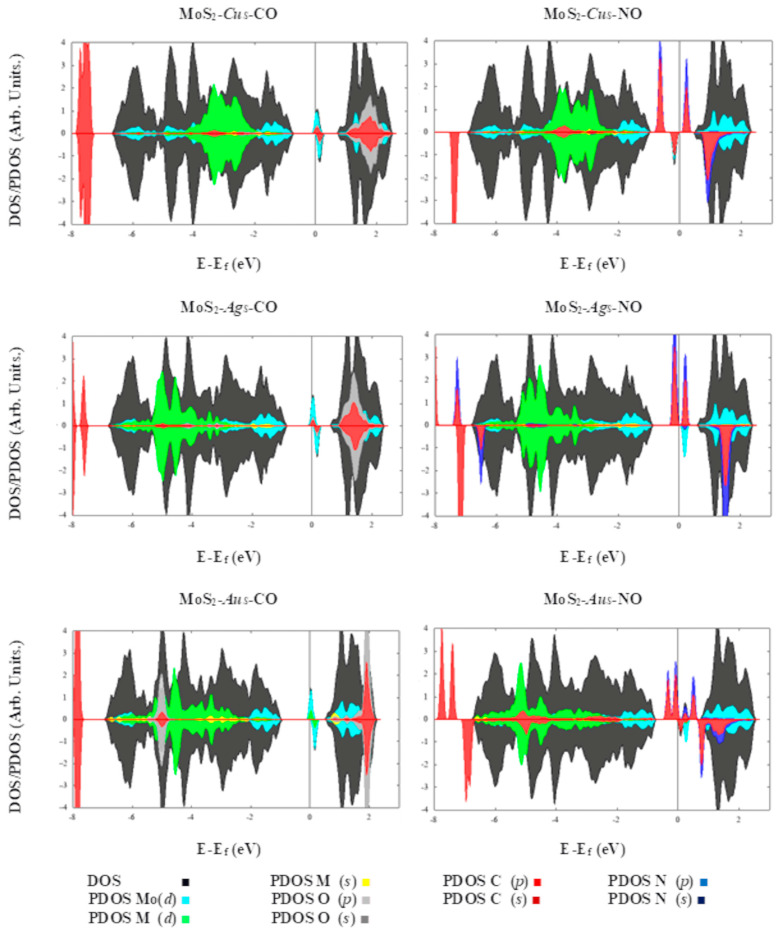
The total (DOS) and projected (PDOS) density of states for the defective systems are shown. PDOS are projected on the annotated orbitals. Fermi energy set at 0 eV.

**Figure 4 ijms-24-10284-f004:**
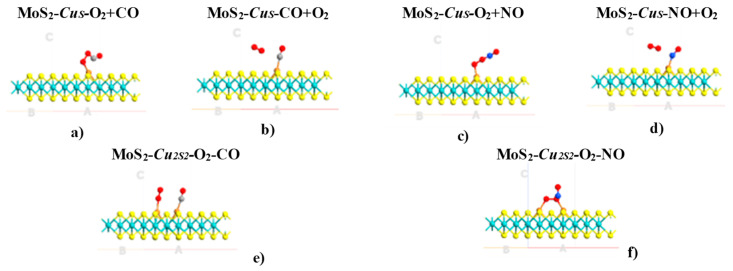
Ground state structures obtained for the defective MoS_2_-*Cu_S_* and MoS_2_-*Cu_2S2_* MLs with previously adsorbed O_2_ molecule (**a**,**c**,**e**,**f**) interacting with CO and NO molecules. Conversely, MoS_2_-*Cu_S_* MLs with CO and NO pollutants previously adsorbed and interacting O_2_ molecule (**b**,**d**) are shown as well. Mo, S, C, N and O atoms are indicated with cyan, yellow, gray, blue and red colors, respectively.

**Figure 5 ijms-24-10284-f005:**
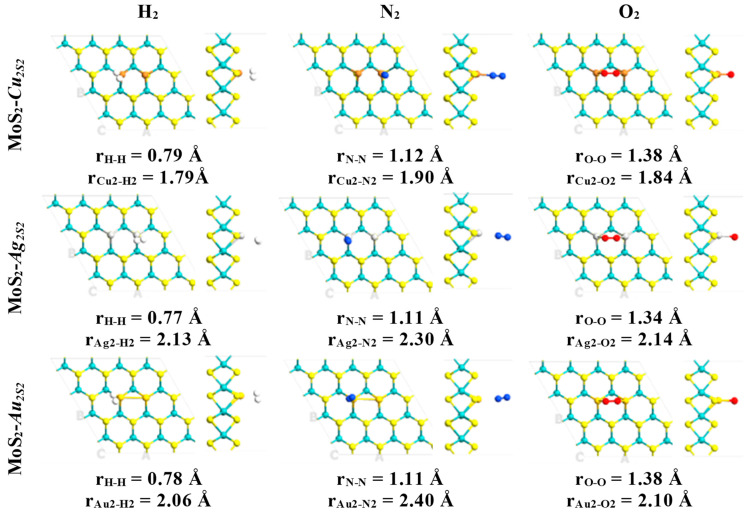
Ground state structures obtained for the defective MoS_2_-*M_2S2_* (M = Cu, Ag and Au) MLs interacting with atmospheric constituents (H_2_, N_2_ and O_2_). Relevant bond lengths are indicated in angstrom units. Mo, S, H, N and O atoms are indicated with cyan, yellow, white, blue and red colors, respectively.

**Figure 6 ijms-24-10284-f006:**
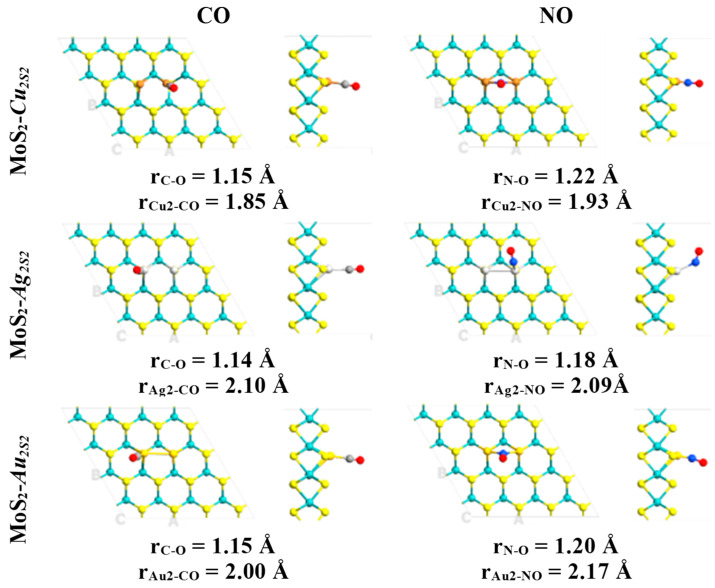
Ground state structures obtained for the defective MLs MoS_2_-*M_2S2_* (M = Cu, Ag and Au) interacting with air pollutants CO and NO. Relevant bond lengths are indicated in angstrom units. Mo, S, C, N and O atoms are indicated with cyan, yellow, gray, blue and red colors, respectively.

**Figure 7 ijms-24-10284-f007:**
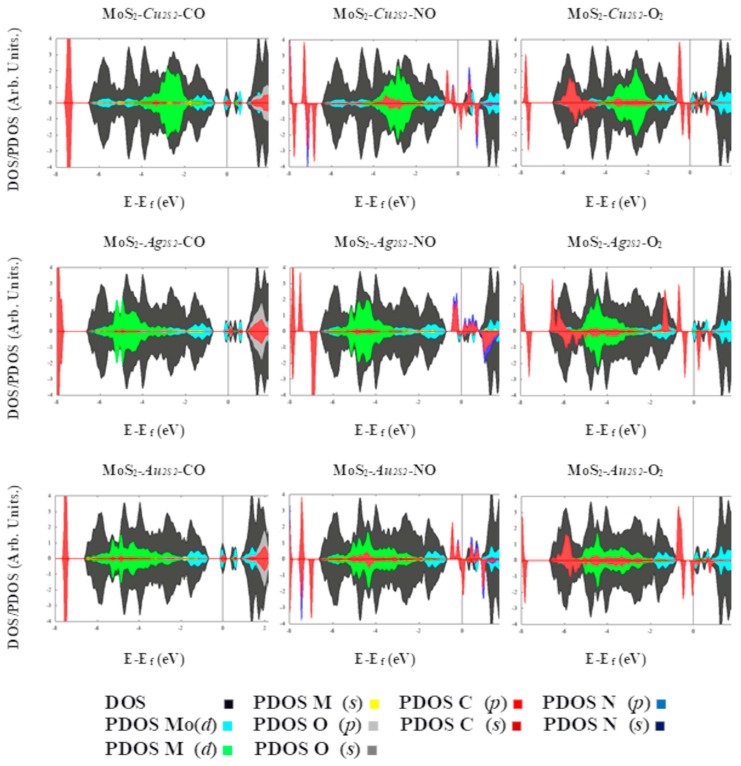
The DOS and PDOS for the defective systems are shown. PDOS are projected on the annotated orbitals. Fermi energy is set at 0 eV.

**Table 1 ijms-24-10284-t001:** Adsorption energy, in kcal·mol^−1^ and eV, of diatomic molecules AB on defective MoS_2_-*M_S_* MLs with a single substitutional atom. Also, total magnetization of the systems, in μ_B_ units, is included. Magnetization on the AB molecule, in μ_B_ units, is split into atom A and B (in parentheses). Total charge transferred to the AB adsorbed molecule, in electron units, as well as its adsorption mode, is annotated.

System	E_ads_ (kcal·mol^−1^)/eV	Total Magnetization (μ_B_)	Magnetization on AB Molecule (μ_B_)	Total Charge on AB Molecule (e)	Adsorption Mode
MoS_2_-*Cu_s_-*H_2_	9.35/0.45	0.67	0.00 (0.00)	0.02	Cu-H_2_
MoS_2_-*Cu_s_-*O_2_	24.88/1.07	0.95	0.34 (0.34)	−0.50	Cu-O_2_
MoS_2_-*Cu_s_-*N_2_	15.21/0.66	0.46	0.00 (0.02)	−0.09	Cu-N-N
MoS_2_-*Cu_s_-*CO	28.54/1.24	0.43	0.01 (0.02)	−0.10	Cu-C-O
MoS_2_-*Cu_s_-*NO	33.18/1.44	2.07	0.47 (0.70)	−0.28	Cu-N-O
MoS_2_-*Ag_s_-*H_2_	6.12/0.27	0.89	0.00 (0.00)	0.05	Ag-H_2_
MoS_2_-*Ag_s_-*O_2_	13.16/0.57	0.94	0.62 (0.64)	−0.42	Ag-O_2_
MoS_2_-*Ag_s_-*N_2_	7.67/0.33	0.82	0.00 (0.01)	−0.02	Ag-N-N
MoS_2_-*Ag_s_-*CO	17.85/0.77	0.69	0.01 (0.02)	−0.01	Ag-C-O
MoS_2_-*Ag_s_-*NO	7.95/0.35	1.75	0.35 (0.88)	−0.19	Ag-O-N
MoS_2_-*Au_s_-*H_2_	5.20/0.23	0.57	0.00 (0.00)	0.06	Au-H_2_
MoS_2_-*Au_s_-*O_2_	11.74/0.51	0.86	0.42 (0.61)	−0.40	Au-O-O
MoS_2_-*Au_s_-*N_2_	6.69/0.29	0.26	0.00 (0.01)	−0.02	Au-N-N
MoS_2_-*Au_s_-*CO	2.95/0.13	0.66	0.00 (0.00)	0.00	Au-O-C
MoS_2_-*Au_s_-*NO	2.48/0.11	1.53	0.44 (0.58)	−0.25	Au-N-O

**Table 2 ijms-24-10284-t002:** Adsorption energy, in kcal·mol^−1^ and eV, of diatomic molecules AB on defective MoS_2_-M_2S2_ MLs with a two substitutional atom. Also, total magnetization of the systems, in μ_B_ units, is included. Magnetization on the AB molecule, in μ_B_ units, is split into atom A and B (in parentheses). Total charge transferred to the AB adsorbed molecule, in electron units, as well as its adsorption mode, is annotated.

System	E_ads_ (kcal·mol^−1^)/eV	Total Magnetization (μ_B_)	Magnetization on AB Molecule (μ_B_)	Total Charge on AB Molecule (e)	Adsorption Mode
MoS_2_-*Cu_2S2_-*H_2_	9.09/0.395	0.03	0.00 (0.00)	0.03	Cu-H_2_
MoS_2_-*Cu_2S2_-*O_2_	38.04/1.65	0.26	0.21 (0.21)	−0.85	Cu-O-O-Cu
MoS_2_-*Cu_2S2_-*N_2_	15.24/0.66	0.04	0.00 (0.01)	−0.12	Cu-N-N
MoS_2_-*Cu_2S2_-*CO	30.21/1.31	0.03	0.00 (0.00)	−0.12	Cu-C-O
MoS_2_-*Cu_2S2_-*NO	38.89/1.69	0.94	0.45 (0.45)	−0.51	Cu-N-O
MoS_2_-*Ag_2S2_-*H_2_	5.306/0.23	0.02	0.00 (0.00)	0.04	Ag-H_2_
MoS_2_-*Ag_2S2_-*O_2_	22.41/0.97	0.94	0.48 (0.48)	−0.60	Ag-O-O-Ag
MoS_2_-*Ag_2S2_-*N_2_	7.35/0.32	0.02	0.00 (0.00)	−0.02	Ag-N-N
MoS_2_-*Ag_2S2_-*CO	18.55/0.80	0.02	0.00 (0.00)	−0.03	Ag-C-O
MoS_2_-*Ag_2S2_-*NO	21.41/0.93	1.15	0.41 (0.66)	−0.25	Ag-N-O
MoS_2_-*Au_2S2_-*H_2_	5.102/0.22	0.03	0.01 (0.01)	0.06	Au-H_2_
MoS_2_-*Au_2S2_-*O_2_	22.078/0.96	0.28	0.25 (0.25)	−0.71	Au-O-O-Au
MoS_2_-*Au_2S2_-*N_2_	5.903/0.26	0.04	0.00 (0.00)	−0.02	Au-N-N
MoS_2_-*Au_2S2_-*CO	24.82/1.08	0.03	0.00 (0.00)	−0.05	Au-C-O
MoS_2_-*Au_2S2_-*NO	25.65/1.11	0.87	0.33 (0.36)	−0.37	Au-N-O

## Data Availability

The data presented in this study are available on request from the corresponding author.

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
