# Peer review of "Substitutional Coinage Metals as Promising Defects for Adsorption and Detection of Gases on MoS_2_ Monolayers: A Computational Approach"

_ijms, 2023, doi:10.3390/ijms241210284_

Round 1
Reviewer 1 Report
Manuscript presented interesting results and my comments are following:
1. Authors should revise the abstract section with more details about the obtained resuls.
2. Please highlight the manuscript novelty/last paragraph of introduction section.
3. More discussion for the obtained results should be provide.
Minor correction
Author Response
Reviewer 1.
Manuscript presented interesting results and my comments are following:
- Authors should revise the abstract section with more details about the obtained results.
- We thank the reviewer’s comment. In consequence, a more detailed abstract has been included in the revised version.
Defective molybdenum disulfide (MoS2) monolayers (MLs) modified with coinage metal atoms (Cu, Ag and Au) embedded in sulfur vacancies are studied at a dispersion-corrected density functional level. Atmospheric constituents (H2, O2 and N2) and air pollutants (CO and NO), known as secondary greenhouse gases, are adsorbed on up to two atoms embedded into sulfur vacancies in MoS2 MLs. The adsorption energies suggest that the NO (1.44 eV) and CO (1.24 eV) are chemisorbed more strongly than O2 (1.07 eV) and N2 (0.66 eV) on the ML with a cooper atom substituting a sulfur atom. Therefore, the adsorption of N2 and O2 does not compete with NO or CO adsorption. Besides, NO adsorbed on embedded Cu creates a new level in the band gap. In addition, it was found that the CO molecule could directly react with the pre-adsorbed O2 molecule on atom Cu, forming the complex OOCO, via Eley–Rideal reaction mechanism. The adsorption energies of CO, NO and O2 on Au2S2, Cu2S2 and Ag2S2 embedded into two sulfur vacancies were competitive. Charge transference occurs from the defective MoS2 ML to the adsorbed molecules, oxidizing the later ones (NO, CO and O2) since act as acceptors. The total and projected density of states reveal that a MoS2 ML modified with copper, gold, and silver dimers could be used to design electronic or magnetic devices for sensing applications in the adsorption of NO, CO, and O2 molecules. Moreover, NO and O2 molecules adsorbed on MoS2-Au2s2 and MoS2-Cu2s2 introduces a transition from metallic to half-metallic behavior for applications in spintronics. These modified monolayers are expected to exhibit chemiresistive behavior, meaning their electrical resistance changes in response to the presence of NO molecules. This property makes them suitable for detecting and measuring NO concentrations. Also, modified materials with half-metal behavior could be beneficial for spintronic devices, particularly those that require spin-polarized currents.
- Please highlight the manuscript novelty/last paragraph of introduction section.
- We agree with this observation. The revised version of the manuscript includes the following additional discussion in the introduction section.
Several devices have already been manufactured to detect gases and semiconductor/chemiresistive sensors have been adopted commercially owing to its low-cost and straightforward fabrication process [29]. Besides, chemiresistive sensors based on tungsten and disulfide, freestanding as well as modified with zinc oxide nanoparticles, were fabricated and reported [30]. These systems showed a significant response to a range of target analytes. The sensing material has an inherent resistance, which can be modulated by the presence or absence of the analyte, leading to the chemiresistive behavior.
More rare and peculiar would be a sensing material that has an inherent transition from metallic to half-metallic behavior, which can be modulated by the presence or absence of molecules. In this manuscript, it is discussed a few findings about half-metallic materials for gas sensing. However, further improvements are needed in the potential manufacturability of sensors based on 2D materials such as molybdenum and tungsten disulfides for their practical applications.
- We thank the reviewer’s comment. In consequence, a more detailed discussion is included. The following paragraphs and augmented tables were added in the revised version. Some additional corrections are highlighted in the document.
Table 1. Adsorption energy, in kcal·mol-1 and eV, of diatomic molecules AB on defective MoS2-MS MLs with a single substitutional atom. Also, total magnetization of the systems, in μB units, are included. Magnetization on AB molecule, in μB units, is split into atom A and B (in parentheses). Total charge transferred to AB adsorbed molecule, in electron units, as well as its adsorption mode is annotated.
|
System |
Eads (kcal·mol-1)/eV |
Total Magnetization (μB) |
Magnetization on AB molecule (μB) |
Total charge on AB molecule (e) |
Adsorption mode |
|
MoS2-Cus-H2 |
9.35/0.45 |
0.67 |
0.00 (0.00) |
0.02 |
Cu-H2 |
|
MoS2-Cus-O2 |
24.88/1.07 |
0.95 |
0.34 (0.34) |
-0.50 |
Cu-O2 |
|
MoS2-Cus-N2 |
15.21/0.66 |
0.46 |
0.00 (0.02) |
-0.09 |
Cu-N-N |
|
MoS2-Cus-CO |
28.54/1.24 |
0.43 |
0.01 (0.02) |
-0.10 |
Cu-C-O |
|
MoS2-Cus-NO |
33.18/1.44 |
2.07 |
0.47 (0.70) |
-0.28 |
Cu-N-O |
|
MoS2-Ags-H2 |
6.12/0.27 |
0.89 |
0.00 (0.00) |
0.05 |
Ag-H2 |
|
MoS2-Ags-O2 |
13.16/0.57 |
0.94 |
0.62 (0.64) |
-0.42 |
Ag-O2 |
|
MoS2-Ags-N2 |
7.67/0.33 |
0.82 |
0.00 (0.01) |
-0.02 |
Ag-N-N |
|
MoS2-Ags-CO |
17.85/0.77 |
0.69 |
0.01 (0.02) |
-0.01 |
Ag-C-O |
|
MoS2-Ags-NO |
7.95/0.35 |
1.75 |
0.35 (0.88) |
-0.19 |
Ag-O-N |
|
MoS2-Aus-H2 |
5.20/0.23 |
0.57 |
0.00 (0.00) |
0.06 |
Au-H2 |
|
MoS2-Aus-O2 |
11.74/0.51 |
0.86 |
0.42 (0.61) |
-0.40 |
Au-O-O |
|
MoS2-Aus-N2 |
6.69/0.29 |
0.26 |
0.00 (0.01) |
-0.02 |
Au-N-N |
|
MoS2-Aus-CO |
2.95/0.13 |
0.66 |
0.00 (0.00) |
0.00 |
Au-O-C |
|
MoS2-Aus-NO |
2.48/0.11 |
1.53 |
0.44 (0.58) |
-0.25 |
Au-N-O |
Table 2. Adsorption energy, in kcal·mol-1 and eV, of diatomic molecules AB on defective MoS2-M2S2 MLs with a two substitutional atom. Also, total magnetization of the systems, in μB units, are included. Magnetization on AB molecule, in μB units, is split into atom A and B (in parentheses). Total charge transferred to AB adsorbed molecule, in electron units, as well as its adsorption mode is annotated.
|
System |
Eads (kcal·mol-1)/eV |
Total Magnetization (μB) |
Magnetization on AB molecule (μB) |
Total charge on AB molecule (e) |
Adsorption mode |
|
|
MoS2-Cu2S2-H2 |
9.09/0.395 |
0.03 |
0.00 (0.00) |
0.03 |
Cu-H2 |
|
|
MoS2-Cu2S2-O2 |
38.04/1.65 |
0.26 |
0.21 (0.21) |
-0.85 |
Cu-O-O-Cu |
|
|
MoS2-Cu2S2-N2 |
15.24/0.66 |
0.04 |
0.00 (0.01) |
-0.12 |
Cu-N-N |
|
|
MoS2-Cu2S2-CO |
30.21/1.31 |
0.03 |
0.00 (0.00) |
-0.12 |
Cu-C-O |
|
|
MoS2-Cu2S2-NO |
38.89/1.69 |
0.94 |
0.45 (0.45) |
-0.51 |
Cu-N-O |
|
|
MoS2-Ag2S2-H2 |
5.306/0.23 |
0.02 |
0.00 (0.00) |
0.04 |
Ag-H2 |
|
|
MoS2-Ag2S2-O2 |
22.41/0.97 |
0.94 |
0.48 (0.48) |
-0.60 |
Ag-O-O-Ag |
|
|
MoS2-Ag2S2-N2 |
7.35/0.32 |
0.02 |
0.00 (0.00) |
-0.02 |
Ag-N-N |
|
|
MoS2-Ag2S2-CO |
18.55/0.80 |
0.02 |
0.00 (0.00) |
-0.03 |
Ag-C-O |
|
|
MoS2-Ag2S2-NO |
21.41/0.93 |
1.15 |
0.41 (0.66) |
-0.25 |
Ag-N-O |
|
|
MoS2-Au2S2-H2 |
5.102/0.22 |
0.03 |
0.01 (0.01) |
0.06 |
Au-H2 |
|
|
MoS2-Au2S2-O2 |
22.078/0.96 |
0.28 |
0.25 (0.25) |
-0.71 |
Au-O-O-Au |
|
|
MoS2-Au2S2-N2 |
5.903/0.26 |
0.04 |
0.00 (0.00) |
-0.02 |
Au-N-N |
|
|
MoS2-Au2S2-CO |
24.82/1.08 |
0.03 |
0.00 (0.00) |
-0.05 |
Au-C-O |
|
|
MoS2-Au2S2-NO |
25.65/1.11 |
0.87 |
0.33 (0.36) |
-0.37 |
Au-N-O |
|
In these cases, CO and NO molecules are attached to the defective monolayer by C and N atoms, correspondingly (Table 1). Interatomic bond lengths for CO and NO on MoS2-Cus are 1.14 and 1.18 Å, respectively. In comparison, in the gas phase, these bond lengths are 1.12 and 1.15 183 Å, respectively.
In these cases, CO and NO molecules are attached to the defective monolayer by C and N atoms, correspondingly (Table 1). Interatomic bond lengths for CO and NO on MoS2-Cus are 1.14 and 1.18 Å, respectively. In comparison, in the gas phase, these bond lengths are 1.12 and 1.15 Å, respectively.
Similarly, adsorbed O2 molecule is obtained with a longer O-O bond length, relative to 1.23 Å computed in the gas phase. The 1.32 and 1.34 Å O-O bond lengths obtained in cases of MoS2-Ags and MoS2-Cus suggest the achievement of the superoxide state in case of oxygen molecule [38]. This interaction leads to atop adsorption modes (Table 1), with the shortest distance between the defect and O2 molecule ranging from 1.97 to 2.27 Å (Figure 1).
The adsorption energies of O2, CO and NO on Ag2S2 and Cu2S2 dimers embedded into sulfur vacancies are similar. In contrast, O2 is the only species that forms bridge adsorption modes with the pairs of embedded metal atoms (Table 2). The other species exhibit atop adsorption modes, firmly attached to a single substitutional defect on the heteroatoms (Table 2).
Thus, the adsorption energies indicate that NO and CO are more strongly chemisorbed than O2 and N2 on the ML when a copper atom substitutes for a sulfur atom (Table 1). The above can be visualized by the shorter bond lengths between NO (1.78 Å)and CO (1.86 Å) molecules and defective surface MoS2-CuS, in comparison with those established with O2 (1.97 Å) and N2 (1.91 Å) (Figure 1 and Figure 2).
Since O2 leads to peculiar behaviors in case of single substitutional defects, the following discussion elucidate its interaction with embedded dimers. First, it is noticed that O2, adsorbed on any dimer under study, achieves the superoxide state. The above can be argued by the O-O bond lengths obtained, ranging from 1.34 to 1.38 Å (Figure 5 and Figure 6) [38]. In addition, the total magnetization of these systems is obtained ranging from 0.26 to 0.94 μB (See Table 2). However, this total magnetization is comparable to the individual contribution of each oxygen atom adsorbed on Au2S2 or Cu2S2 defects (Table 2). Since this fact sounds a little bit contradictory, spin density isosurfaces were obtained (See Supporting Information). It is noticed that uniquely in case of O2, obtained with important spin-up contributions, total magnetization is reduced by the introduction of spin-down electrons on Mo atoms located behind the embedded defect. It is relevant to note that the Mo layer acts as an electron reservoir, transferring a certain amount of charge to the adsorbed molecule through the metal defect. Thus, resembles a parallel plate capacitor discharging into the adsorbed molecule as resistive component.
As discussed below, the charge transferred from the ML to the O2 molecule induced its superoxide state. Admirably, at the same time, antiparallel electrons are found surrounding Mo atoms near Au2S2 or Cu2S2 defects. The above is consistent with recent reports on the formation of reactive oxygen species on nanostructured MoS2 derivatives [42,43].
Overall, MoS2-MS and MoS2-M2S2 systems possess intrinsic electrons and holes at a temperature greater than absolute zero. In the absence of an interacting gas, such as O2, CO or NO, the sensor would ideally show a given conductivity and resistivity. However, the charge distribution in the sensing material could change when it interacts with an external gas molecule, and the modification depends upon the strength of the molecule-ML chemical interaction. Strong interactions between NO and O2 adsorbed on MoS2-CuS system is attributed to the charge transfer and orbital hybridization between the molecules and the embedded CuS atom. Thus, MoS2-CuS, proposed as chemiresistive sensing material, showed a significantly higher response to NO. Upon adsorption, each spin-polarized 2π* level of the NO is splitting into two states and the spin up 2π* the level is shifted below the Fermi level due to the hybridization between Cu (d) and NO (2π*) orbital.
On the other hand, a more peculiar behavior is showed by MoS2-Au2S2 and MoS2-Cu2S2 sensing materials. The above due to their transition from metallic to half-metallic materials, which can be modulated by the presence or absence of O2 and NO. Once again, spin-up 2π* orbitals, for NO or O2, are shifted below the Fermi level due to the hybridization between d metal orbitals and 2π* ones of the interacting molecules, thus an energy gap is opened.
Reviewer 2 Report
In the current study, the Authors reported on the application of a promising process based on the adsorption of diatomic molecules on group 11 atoms embedded into mono- and di-vacancies of sulfur on molybdenum disulfide monolayer. Collectively, the study is solid, attractive, and valuable for the scientific community. The results were clearly depicted, as well as the computational approach. Overall, the work appears to be almost ready for acceptance. Anyway, I recommend accepting it after minor revisions.
The following few comments are reported:
1. Line 30: The acronymous for molybdenum disulfide monolayer should appear here.
2. What about the process’s overall costs? Please consider adding an economic feasibility of the process, if possible. This information could be useful to evaluate the real application of the study.
3. Conclusions: Rather than summarizing the results section, the Conclusion section should conclude what this means for the overall process and its application in the real world. Please, consider clarifying how the current research can be useful for real applications.
Author Response
Reviewer 2.
In the current study, the Authors reported on the application of a promising process based on the adsorption of diatomic molecules on group 11 atoms embedded into mono- and di-vacancies of sulfur on molybdenum disulfide monolayer. Collectively, the study is solid, attractive, and valuable for the scientific community. The results were clearly depicted, as well as the computational approach. Overall, the work appears to be almost ready for acceptance. Anyway, I recommend accepting it after minor revisions.
The following few comments are reported:
- Line 30: The acronymous for molybdenum disulfide monolayer should appear here.
- We thank this observation. The following phrase appears in the revised manuscript.
The potential applications, such as optoelectronics, of molybdenum disulfide (MoS2) monolayer (ML) have been widely studied due to its optical gap ranging from 1.8 to 1.9 eV [1], high thermal stability [1],…
- What about the process’s overall costs? Please consider adding an economic feasibility of the process, if possible. This information could be useful to evaluate the real application of the study.
- We thank the reviewer’s observation. In order to clearly stablish the feasibility of the materials proposed in this work, some comments about devices and applications based on MoS2 monolayers or similar systems were included. In brief, chemical vapor deposition, sonication, and even mechanical exfoliation can be applied to obtain the monolayers, of course with a variable quality. In addition, laser ablation and plasma etching are useful techniques to create sulfur vacancies. Finally, the molecules under study are natural atmospheric components and gases obtained from common combustion engines.
https://doi.org/10.1063/5.0076711
https://doi.org/10.1002/inf2.12161
Several devices have already been manufactured to detect gases and semiconductor/chemiresistive sensors have been adopted commercially owing to its low-cost and straightforward fabrication process [29]. Besides, chemiresistive sensors based on tungsten and disulfide, freestanding as well as modified with zinc oxide nanoparticles, were fabricated and reported [30]. These systems showed a significant response to a range of target analytes. The sensing material has an inherent resistance, which can be modulated by the presence or absence of the analyte, leading to the chemiresistive behavior. In addition, hydrodesulfurization process, crucial to remove sulfur from petroleum and to produce ultraclean fuels, commonly employs derivatives of bulk MoS2 as catalyst [31,32]. Recently, the interest of petrochemical industry on materials, with improved performance, has attracted the attention on nanostructured forms of MoS2, such as nanoparticles,[33] nanorods and nanosheets [34,35].
These materials based on nanostructured MoS2 are not only relevant for sensing applications [7,26,36,37], but also hold potential as catalysts for petrochemical applications. Utilizing nanostructured forms of MoS2 is a feasible approach considering the widely used applications of its common bulk derivatives.
- Conclusions: Rather than summarizing the results section, the Conclusion section should conclude what this means for the overall process and its application in the real world. Please, consider clarifying how the current research can be useful for real applications.
We thank the reviewer’s comment. Since our study is theoretical and focused on the basic properties of modified materials with mono- and di- embedded coinage atoms interacting with air components, reported by the first time, a few comments about potential applications were introduced in Conclusions section. Some comments of our previous Conclusions were omitted.
On the other hand, the adsorption of NO molecules onto MoS2-Cu2S2 induces a transition from metallic to semiconductor behavior, for spin-up-down electrons. Thus, this material is interesting for spintronic applications.
For the first time, the electronic behavior of some diatomic molecules adsorbed on Au2S2, Cu2S2 and Ag2S2 embedded into sulfur di-vacancies was studied as well. The introduction of the diatomic molecules affects mostly electronic and magnetic properties.
The changes induced by the adsorbed molecules are selective, meaning they depend on the specific chemical species involved. This observation suggests potential applications for the modified MLs as gas phase molecule sensors.
Sensors based on modified MoS2 monolayers, due to their expected chemiresistive behavior, can be proposed for NO molecules. Furthermore, the introduction of certain gas molecules has been found to modify the magnetic properties of the materials, resulting in a new category of sensitive materials. Additionally, the modified materials exhibit an intriguing phenomenon known as half-metal behavior, which holds potential for spintronic devices that rely on spin-polarized currents.
Reviewer 3 Report
Dear authors,
thank you for your work. Please, write more keywords (8-10).
Please, made "Materials and methods" part. ("Theoretical method" is short and need more details about modeling and experimenlat data etc)
Best regards, reviewer
Author Response
Reviewer 3.
Dear authors,
thank you for your work. Please, write more keywords (8-10).
- We included a few additional keywords as follows:
Keywords: molybdenum disulfide monolayer; molecular adsorption; half-metallic behavior; greenhouse gases; chemiresistive sensors, magnetic sensors, density functional theory; coinage metal defects.
Please, made "Materials and methods" part. ("Theoretical method" is short and need more details about modeling and experimenlat data etc).
- We thank this observation. Consequently, the methods section was improved with additional details as follows:
Theoretical Method
Dispersion-corrected density functional method, labeled as PBE-D3(BJ)/USP, was previously used since obtained reliable results for defective MoS2 ML with substitional coinage metals [16,17]. The quantum chemistry package Quantum Espresso 6.1 [44] was used for most of the calculations. The full methodology Ultra-soft pseudopotentials, obtained by the Rappe Rabe Kaxiras Joannopoulos method, were used for the interacting molecules as well. Furthermore, in previous studies, it was demonstrated that the 4 × 4 × 1 supercell is sufficiently large to accommodate the molecules under investigation (H2, N2, O2, NO and CO). Previously, a 5 × 5 × 1 supercell was tested at the PBE-D2/USP level [16], obtained no relevant geometrical differences in comparison to the smaller one.
The energy was converged up to 1.0 × 10−7 a.u. with an energy cut-off of 60 Ry. The convergence criteria for the geometry optimization considered a threshold on the total energy of 1.0 × 10−5 a.u., and 1.0 × 10−4 a.u., for the residual forces on each atom. Besides, the BFGS quasi-Newton algorithm was used for structural relaxation. The Brillouin zone was sampled trough a 4 × 4 × 1 k-point grid. The mesh was increased up to 16 × 16 × 1 for the density of states and the projected density of states calculations.
The lattice parameter obtained, after a fully relaxation of the unit cell, was 3.199 Å. This value is pretty similar to the 3.169 Å value measured for the bulk phase MoS2 [45]. Another value of 3.16 Å was obtained, via scanning tunneling microscopy, on MoS2 monolayers on highly ordered pyrolytic graphite [46].
On the other hand, a defect Xy refer to X species, such as metal atoms Mn (n = 1, 2) or vacancies Vn (n = 1, 2 ), occupying the site of the Y species into MoS2 ML; sulfur atoms Sn (n = 1, 2) in the current case. Multiple sulfur vacancies are denoted as VSn (n = 1, 2). Embedded clusters MnSn (n = 1, 2, M = Cu, Ag or Au) occupying sulfur vacancies were considered. The adsorption energies of the metal atoms embedded into sulfur vacancies and the adsorption energies of the diatomic molecules on metal were calculated as follow:
EMetal-ads= E(MoS2-VnSn) + E(Mn) – E(MoS2-MnSn) (1)
EAB-ads= E(MoS2-MnSn) + E(AB) – E(MoS2-MnSn-AB) (2)
EAB-AB-ads= E(MoS2-MnSn+AB) + E(AB) – E(MoS2-MnSn-AB-AB) (3)
E(MoS2-VnSn+AB), E(MoS2-MnSn) and E(MoS2-MnSn-AB) are the total energies of defective MLs: with n sulfur vacancies VnSn, with embedded clusters MnSn (n = 1, 2, M = Cu, Ag or Au) occupying sulfur vacancies, and interacting with AB diatomic molecules, respectively. Structures with two metal atoms to form the defects MnSn (n= 1-2) were studied. Total electrostatic potential (ESP) was mapped on isosurfaces with 0.01 a.u. of electron density. Bader’s charge analyses were calculated by Henkelman’s group software [47]. Similarly, individual contributions of the A and B atoms, forming the AB molecule, to the total magnetization were computed by the Bader’s procedure and the spin density.
Reviewer 4 Report
Paper devoted to investigation of substitutional coinage metals as promising defects for adsorption and detection of gases on MoS2 monolayers. This topic interesting and relevant. But there are some comments:
1. There is no Methods section in the paper.
2. I think that it is better to divide the Conclusions into several points.
Author Response
Reviewer 4.
Paper devoted to investigation of substitutional coinage metals as promising defects for adsorption and detection of gases on MoS2 monolayers. This topic interesting and relevant. But there are some comments:
- There is no Methods section in the paper.
- A methods section is already included in the original version of our manuscript. Below we include the revised version of this section.
- I think that it is better to divide the Conclusions into several points.
- We reformed the Conclusions section with a few comments about applications and divided in several points as follows:
Chemical activation of the molybdenum disulfide ML was achieved through coinage metal atoms (Cu, Ag, and Au) embedded into mono- and di-sulfur vacancies VS, and V2S2, respectively.
Atmospheric components (N2, O2, H2) and air pollutants known as secondary greenhouse gases (CO and NO) can be attached to the surface since their adsorptions are energetically favorable in most cases. Among them, CO and NO on the substitutional copper atom obtained higher adsorption energies. Furthermore, N2 and O2 adsorption could not compete with NO or CO adsorption at all.
The enlargement of bond lengths observed after adsorption serves as preliminary evidence of bond weakening due to charge transfer from the copper atom embedded in the sulfur vacancies to the CO and NO molecules.
The adsorption of the CO, N2 and H2 molecules do not lead to changes in the electronic properties of the defective MoS2 MLs. In contrast, as evidenced by total and projected density of states, NO adsorbed on embedded Cu creates a new level in the band gap of the MoS2 ML.
In addition, it was found that the CO molecule could directly react with the pre-adsorbed O2 molecule on atom Cu, forming the complex OOCO, via Eley–Rideal reaction mechanism.
The adsorption energies of CO, NO and O2 on Au2S2, Cu2S2 and Ag2S2 dimers embedded into sulfur di-vacancies were similar. Charge transfer occurs from the modified MLs to the adsorbed molecules, oxidizing NO, CO and O2 and acting as acceptors.
In particular, NO and O2 molecules adsorbed on MoS2-Au2S2 and MoS2-Cu2S2 introduces a transition from metallic to half-metallic behavior. Interestingly, CO and NO molecules do not react with the pre-adsorbed O2 molecule. As a result, it does not displace it.
On the other hand, the adsorption of NO molecules onto MoS2-Cu2S2 induces a transition from metallic to semiconductor behavior, but only for spin-up-down electrons. Thus, this material is interesting for spintronic applications.
For the first time, the electronic behavior of some diatomic molecules adsorbed on Au2S2, Cu2S2 and Ag2S2 embedded into sulfur di-vacancies was studied as well. The introduction of the diatomic molecules affects mostly electronic and magnetic properties.
Changes introduced by the adsorbed molecules are selective, i.e., depending on the chemical species involved, suggesting potential applications of the modified MLs as sensors of molecules in the gas phase.
Sensors based on modified MoS2 monolayers, due to their expected chemiresistive behavior, can be proposed for NO molecules. Moreover, the magnetic properties were modified by the introduction of some gas molecules, leading to another category of sensitive materials. In addition, another intriguing phenomenon emerged from the modified materials, the half-metal behavior, could be useful for spintronic devices in which spin-polarized currents are crucial.
Round 2
Reviewer 2 Report
The authors have satisfactorily replied to the reviewer's comments. I recommend accepting the manuscript in this last form.